# Internet-of-Things (IoT) Platform for Road Energy Efficiency Monitoring

**DOI:** 10.3390/s23052756

**Published:** 2023-03-02

**Authors:** Asmus Skar, Anders Vestergaard, Shahrzad M. Pour, Matteo Pettinari

**Affiliations:** 1Environmental and Resource Engineering, Technical University of Denmark, Nordvej, B119, 2800 Kongens Lyngby, Denmark; 2Applied Mathematics and Computer Science, Technical University of Denmark, Richard Petersens Plads, 2800 Kongens Lyngby, Denmark; 3Danish Road Directorate, Guldalderen 12, 2640 Hedehusene, Denmark

**Keywords:** live road condition assessment, infrastructure monitoring, pavement analysis, road energy labeling, emission, smart cities

## Abstract

The road transportation sector is a dominant and growing energy consumer. Although investigations to quantify the road infrastructure’s impact on energy consumption have been carried out, there are currently no standard methods to measure or label the energy efficiency of road networks. Consequently, road agencies and operators are limited to restricted types of data when managing the road network. Moreover, initiatives meant to reduce energy consumption cannot be measured and quantified. This work is, therefore, motivated by the desire to provide road agencies with a road energy efficiency monitoring concept that can provide frequent measurements over large areas across all weather conditions. The proposed system is based on measurements from in-vehicle sensors. The measurements are collected onboard with an Internet-of-Things (IoT) device, then transmitted periodically before being processed, normalized, and saved in a database. The normalization procedure involves modeling the vehicle’s primary driving resistances in the driving direction. It is hypothesized that the energy remaining after normalization holds information about wind conditions, vehicle-related inefficiencies, and the physical condition of the road. The new method was first validated utilizing a limited dataset of vehicles driving at a constant speed on a short highway section. Next, the method was applied to data obtained from ten nominally identical electric cars driven over highways and urban roads. The normalized energy was compared with road roughness measurements collected by a standard road profilometer. The average measured energy consumption was 1.55 Wh per 10 m. The average normalized energy consumption was 0.13 and 0.37 Wh per 10 m for highways and urban roads, respectively. A correlation analysis showed that normalized energy consumption was positively correlated to road roughness. The average Pearson correlation coefficient was 0.88 for aggregated data and 0.32 and 0.39 for 1000-m road sections on highways and urban roads, respectively. An increase in IRI of 1 m/km resulted in a 3.4% increase in normalized energy consumption. The results show that the normalized energy holds information about the road roughness. Thus, considering the emergence of connected vehicle technologies, the method seems promising and can potentially be used as a platform for future large-scale road energy efficiency monitoring.

## 1. Introduction

The road transportation sector is a dominant and growing energy consumer. Data from the International Energy Agency (IEA) show that the road transport sector accounts for 24% of all CO_2_ emissions from energy, of which, 45.1% comes from the passenger transport [1]. Furthermore, it is the only sector where emissions have increased in recent years [2]. With the reduction of CO_2_ emissions at the forefront of European policy, the road transport sector is focusing on electrification, e.g., by promoting electric vehicles (EVs), and improving the energy efficiency of the transport infrastructure.

In order to accelerate the shift towards a more energy-saving road infrastructure, the tire industry has implemented labeling that primarily highlights fuel efficiency [3,4,5]. For many years, the car industry has also had a similar labeling policy displaying fuel efficiency and projected CO_2_ emissions [6]. Other examples of energy labeling in the civil infrastructure domain include the building sector [7], and the railway sector [8]. The purpose of energy labeling is threefold: (i) to increase consumer and user awareness; (ii) to help consumers/users save money by choosing products that consume less energy and thereby emit less CO_2_; and (iii) to encourage manufacturers to develop these energy-saving products and reduce energy consumption.

Initiatives to quantify the road infrastructure impact on energy consumption have focused on measuring/modeling rolling resistance [9,10,11,12,13]. This has led to the development of energy-saving products such as asphalt courses with low-rolling resistance properties [14,15]. However, from a road operator’s perspective, there is currently no standard method for monitoring or labeling the energy efficiency of road pavements. This results in the following implications: (i) road owners and operators are limited to restricted types of data when managing their road networks, optimizing the timing of repair efforts and prioritizing resources; (ii) the effect of initiatives to reduce the energy consumption of road networks cannot be measured nor quantified in a uniform and rational manner; (iii) the environmental impacts of road networks are difficult to assess (i.e., from a life cycle perspective) due to the lack of data in the use phase [16].

Smart transportation and smart city traffic management are revolutionizing how cities approach mobility and emergency response while reducing congestion on city streets [17,18,19]. This trend is facilitated by the increasing use of Internet-of-Things (IoT) devices [20] and 5G communication technology [21,22]. The former provides inexpensive sensors and controllers that can be embedded into nearly any physical machine to be controlled and managed remotely. The latter provides the high-speed communications needed for managing and controlling transportation systems in real time with minimal latency. Internet of Vehicles (IoV) technology enables internet connectivity and communication between vehicles and other devices on the network and is an integral part of IoT [23]. Advanced sensors in modern cars enable vehicles to sense, communicate, report, and react to the surrounding environment to benefit drivers, commuters, other vehicles, and authorities [24].

Considering the massive amount of real-time data gathered by vehicles, which is in the order of a few gigabytes per hour per vehicle [25], research in this area has been focusing on the development of computing environments to reduce pressure on cloud servers [23,24]. Another challenge relates to security and privacy aspects; data generated by vehicles is transmitted on public servers, and therefore sensitive information can easily be intercepted and tampered with [23]. As a result, developing secure data exchange between vehicles and infrastructure [23,26] is another critical building block to fully utilizing IoV technology.

The global increase of EVs [27], combined with advancements in IoT and IoV technology, creates new opportunities for road usage insights and optimization. Vehicle sensor readings are directly or indirectly related to the ride-surface conditions over which the vehicle is passing. If sensor readings can be collected, a fleet of modern cars can provide valuable input to augment and enrich the data needs for pavement management, and operation [28].

This work is motivated by the need for a road energy efficiency monitoring concept that can provide frequent measurements over large areas across all weather conditions. The idea advocated herein is to utilize the readings of in-vehicle sensors. The sensors can be installed during car manufacturing or retrofitted by the owners after purchase. Unlike previous work on energy consumption measurements from in-vehicle sensors that focused on road geometry [29,30,31] and driving behavior [31,32], this contribution investigates the link between energy consumption and the physical condition of the road pavement.

The main contributions of this study are: (i) the presentation of an IoT platform for performing road energy efficiency monitoring suitable for wide-area implementation; (ii) a normalization technique of energy data taking into account the effects of road slope, vehicle speed, and acceleration; (iii) an investigation of the link between vehicle energy consumption and road pavement roughness; and (iv) a demonstration of the new method over real data obtained by cars driving over a variety of road types.

The paper commences with a review of related research that focuses on the road roughness influence on vehicle energy consumption. The review is followed by stating the specific study objectives and contributions. Next, a framework and methodology for developing an IoT platform for road energy efficiency monitoring are presented. This is followed by an experimental study and validation of the proposed method. Next, the link between vehicle energy consumption and road roughness is investigated. Finally, the method is demonstrated on data obtained from ten nominally identical electric cars driven over highways and urban roads.

## 2. Related Research Work

The road surface characteristics affect the pavement-–vehicle interaction and contribute to vehicle operating costs [33]. When a vehicle travels at constant speed on an uneven road pavement surface, the mechanical work dissipated in the vehicle’s suspension system is compensated by engine power and thus contributes to energy consumption.

The most popular model for evaluation of vehicle operating costs, including energy/fuel consumption, is the Highway Developmental Management System, Version 4 (HDM-4) model (see, e.g., [34]). The HDM-4 model relates to the International Roughness Index (IRI) [35] to fuel consumption utilizing empirical relationships [36,37,38,39,40]. The empirical models have gained importance in life-cycle assessment (LCA) and the relation between pavement condition, energy consumption, and environmental impacts [39,41].

In the work of [37], the HDM-4 model was used to study the vehicle operating costs and vehicle fuel consumption caused by pavement conditions. The model was calibrated using field data and found a linear relationship between changes in fuel consumption and road roughness. The results showed that an increase in IRI of 1 m/km resulted in 2–3% increase in fuel consumption of passenger cars regardless of speed.

Related studies by [38,39] utilized the HDM-4 model and MOVES (MOtor Vehicle Emission Simulator) [42] for predicting impacts on estimates of energy consumption during the use phase. In [38], the authors reported that ignoring IRI variation or traffic congestion could lead to underestimations in energy consumption by approximately 6%. In [39], the authors found that vehicle efficiency accounts for about 27% of the potential total energy savings, and potential savings from pavement roughness can be up to 7%

To develop a more direct link between energy dissipation and pavement roughness a mechanistic model approach was proposed in [43,44]. The authors used the quarter-car model [35] to evaluate the effect of pavement conditions on vehicle fuel consumption. In addition, the authors presented a method for calibrating the model with measured data. The dissipated energy was shown to scale with the mean square of suspension motion, yielding a quadratic increase in fuel consumption with increasing IRI. Monte Carlo simulations were used to estimate the sensitivity of roughness-induced excess fuel consumption to IRI. A Spearman’s rank correlation test showed that energy consumption was positively correlated to pavement roughness with correlation coefficients in the range of 0.45–0.66 for the five vehicle types considered. Monte Carlo simulations were also used to estimate the sensitivity of a mechanical energy dissipation model in [45]. The authors reported a Spearman’s rank correlation of 0.49 and 0.47 for sedan cars and trucks (respectively).

Related studies by [46,47] proposed more computationally-efficient formulations for determining the energy dissipated in the vehicle suspension system. Both studies found a quadratic relationship between changes in fuel consumption and roughness. In the work of [48], a combined roughness and displacement energy dissipation model was utilized, showing that the influence of concrete pavement stiffness on energy consumption is negligible. Similar findings are reported for asphalt roads [49].

A data-driven approach to study the impact of road geometry and gravel pavement roughness on the fuel consumption of logging trucks was proposed in the work of [50]. The truck’s CAN-bus system was utilized to log fuel consumption data, and the road geometry data were measured with a profilograph. An initial Pearson correlation test of 1000-m road sections showed that fuel consumption was negatively correlated to truck speed (−0.52) and positively correlated to gradient (0.68) and pavement roughness (0.37). In all cases, the results were statistically significant. The study did not consider the influence of accelerations (although this was identified as an essential factor), nor did the authors explain the relatively strong negative correlation between speed and energy consumption.

A summary of related research w.r.t. modeling and measurements of the pavement roughness influence on vehicle energy consumption is listed in Table 1; the Table shows the method/approach used and its main limitations.

## 3. Objectives and Methodology

While some of the methods identified in the technical literature have merit, there are also drawbacks: (i) models are relatively complex and require calibration; (ii) the mechanical model used needs extensive system characterization; (iii) models have been developed in a simulation setting and are, therefore, dependent on or limited by the complexity of the simulation, calibration procedures, as well as model correctness; (iv) the experimental design utilized is limited to a specific vehicle or pavement type; and (v) the overall scalability of the methods is not addressed, e.g., how these models should be implemented in practice, making the methodology suitable for wide-area implementation.

The objective of the current work is threefold: (i) develop a practical method for road energy efficiency monitoring based on in-vehicle sensor data; (ii) establish a link between vehicle energy consumption and road pavement roughness; and (iii) demonstrate and validate the new method over real data obtained by cars driving over a variety of road types.

Specifically, the proposed system is based on measurements from in-vehicle sensors. The measurements are collected onboard with an IoT device and then periodically transmitted before being processed, normalized, and saved in a database. The normalization procedure involves modeling the vehicle’s driving resistances in the driving direction. It is hypothesized that the energy remaining after normalization holds information about wind conditions, vehicle-related inefficiencies, and, lastly, the physical condition of the road pavement. Hence, the method addresses all the shortcomings mentioned above by utilizing data collected by regular cars in a real-world setting.

## 4. Road Energy Monitoring Concept

### 4.1. Framework

The proposed framework for monitoring road energy efficiency is schematically presented in Figure 1. As can be seen, the scheme is composed of five elements: (i) data collection from a fleet of electric vehicles (EVs)—in-vehicle sensor data are collected and synchronized automatically with a computer connected to the vehicle’s controller area network (CAN) bus; (ii) transmission and storage—the data are transmitted to a cloud-based system before being stored in a database for further processing and analysis; (iii) modeling—normalization of data utilizing a physical model; (iv) data aggregation—averaging of data over several vehicle passes and a comparison of normalized vehicle energy consumption and standard road roughness measurements, and (v) demonstration—mapping and visualization of energy data.

The overall concept relies on the fact that the instant traction force is measured by a fleet of uniform EVs (i.e., data collection platform). Moreover, measured by EVs are longitudinal acceleration, road slope, and speed. The latter information enables estimation of the main driving resistance forces, i.e., by utilizing a longitudinal vehicle dynamics model (see e.g., [51]), and thus the EVs energy requirements. Therefore, it is hypothesized that the traction forces remaining after normalization (i.e., after subtracting the main driving resistance forces from the measured traction force) hold information about wind conditions, vehicle-related inefficiencies, and the physical condition of the road pavement surface. The latter component is then isolated to quantify road energy efficiency. As an initial attempt to link energy consumption and the physical condition of the road pavement, the normalized energy is compared to the pavement roughness, measured with standard laser-based methods (see, e.g., [52,53]).

### 4.2. Data Collection, Transmission, and Storage

The field measurements presented hereafter are part of the live road assessment (LiRA) project [28,54,55]. This project collected sensor data from a fleet of Renault Zoe EVs operated by Green Mobility (GM), a car-sharing service company. Relevant EV specifications are shown in Table 2.

The GM cars are each retrofitted with an IoT hardware dongle—AutoPi Telematics Unit (3rd generation). This unit includes a single-board Raspberry Pi computer with added GPS and accelerometer modules. The AutoPi units are physically fixed to the frame in the middle of the car. The installation of the devices is depicted in Figure 2. The device is located close to the front axle on the vehicle’s passenger side, inside the middle console.

Overall, an automated data collection process consists of steps of actual data acquisition, data collection, processing, and storage in a high-level definition. Data acquisition is processing data onboard, which involves converting bits to decimals and translating decimal data into actual physical units. Once data are digitized and acquired, they will be structured into unified files in a Jsonlines format, each so-called ‘trip’ data. A ‘trip’ data comprises all measurements from when a car is started until it is turned off. This provides additional metadata such as start and stop position, start and end time, traveled distance, trip identification, etc. The trip data are initially transmitted to a cloud space in GreenMobility servers. This cloud space is considered a temporary solution to be removed once the system is up-scaled and taken to production. Once the data are ready to be exported, it notifies the LiRA big-data pipeline to collect the trip data.

The big-data pipeline [57] plays a coordination role among various steps of an automated data collection module. Such coordination is fulfilled via two aspects: (i) a modularized software architecture, where operational steps are designed and developed in separate modules. (ii) an event-driven software architecture that facilitates online data streaming among the modules. The pipeline is deployed in a Linux-based infrastructure and centrally managed and accessible via a Secure Sockets Layer (SSL) connection.

The AutoPi was configured to collect the raw stream from approximately 50 CAN signals. The present study utilized six CAN signals, i.e., longitudinal acceleration, vehicle speed, wheel torque, traction power, and trip consumption. Moreover, utilized were vehicle location and accelerometer signals collected with the AutoPi. Sensor specifications are presented in Table 3.

### 4.3. Modeling Energy Consumption

This section presents the longitudinal vehicle dynamics model for normalizing energy data. Driving resistances are typically divided into two types: steady-state resistances and dynamic resistances. Steady-state resistance occurs when a vehicle is traveling at a constant speed. Rolling resistance force, aerodynamic drag force, and climbing force resistance all fall into this category. When the vehicle accelerates, a dynamic resistance force occurs. Steady-state resistances continue to act when the vehicle is accelerating.

Newton’s 2nd law can be utilized to formulate a simple 1D representation of the vehicle’s longitudinal dynamics
(1)$$F_{tr}^{p}=F^{D}+F^{R}+F^{\beta }+F^{a}$$
where $F^{D}$ is the aerodynamic drag traction force, $F^{R}$ is the total rolling traction force, $F^{\beta }$ is the climbing traction force, and $F^{a}$ is the inertial traction force (all in units of newton). $F_{tr}^{p}$ is the total predicted traction force, i.e., the force generated by the vehicle motor to overcome the resistive forces. The individual components of Equation (Equation 1) are given below.

The basic formula for the calculation of aerodynamic drag force is given as
(2)$$F^{D}=\mathrm{sign}\left(v+v_{wind}\right)\frac{1}{2}\rho AC_{d}{\left(v+v_{wind}\right)}^{2}$$
where *v* is the speed of the vehicle (meters per second) in the longitudinal direction of the road, *A* is the front area of the vehicle (square meters), ρ is the air density of dry air (kilogram per cubic meter), $C_{d}$ is the aerodynamic drag coefficient (dimensionless) and $v_{wind}$ is the headwind speed (meters per second). The wind speed is negligible compared to the vehicle speed and therefore often omitted. Relevant coefficients can be found in Table 2.

Most rolling resistance calculations assume that a vehicle is driven in a straight line on a dry road surface. Under these conditions, the total rolling resistance $F^{R}$ can be considered equal to the tire rolling resistance force $F^{T,R}$ [51]. This assumption can be made if the road does not undergo plastic deformation, the bearing friction is comparatively small, and the wheel can roll freely with no camber or toe angle. In this specific case, the rolling resistance force may be predicted as
(3)$$F^{R}=mg\cos\theta k^{R,T}\;\mathrm{where}\;k^{R,T}=0.01\left(1+\frac{3.6v}{100}\right)$$
where $k^{R,T}$ is a dimensionless tire rolling resistance coefficient (see e.g., [32]).

The climbing force due to a slope/incline of the road is given as
(4)$$F^{\beta }=mg\sin\theta \approx mg\beta$$
where *m* is the mass of the vehicle (kilogram), *g* is the gravitational acceleration (kilogram per square second), and θ the road slope angle (radians), also referred to as the pitch angle. The road slope is defined as the quotient of its projected vertical and horizontal components and is often given as $\beta =\tan^{-1}\left(\theta \right)=dh/dx$ (percentage), where *h* (meters) is the road altitude. Assuming θ is small compared to unity sinβ≈tanβ≈β.

In order to alter the state of motion of a vehicle with a total mass of *m* from an initial velocity $v_{1}$ to a desired velocity $v_{2}$ with an acceleration *a*, an inertial resistance force must be overcome, given as
(5)$$F^{a}=ma$$
where *a* is the longitudinal acceleration (meters per square second) of the vehicle.

The normalized traction force utilized to rate the energy efficiency can then be estimated from
(6)$$F_{tr}^{n}=F_{tr}^{m}-F_{tr}^{p}=\frac{T_{whl}}{r_{t}}-F_{tr}^{p}$$
where $F_{tr}^{m}$ is the measured traction force, $T_{whl}$ is the measured wheel torque (newton meters) and $r_{t}$ is the radius of the tire (meters).

The traction energy consumed (in units of newton meters) over a distance traveled of Δx is given by
(7)$$E_{\Delta x}=F_{tr}\Delta x$$

An overview of the relevant force and energy components is given in Table 4.

## 5. Vehicle Energy Dataset

### 5.1. Validation of Sensor Data

Before further processing and investigation, the data collection platform was validated. The validation is visualized in Figure 3. The figure compares the CAN sensor reading versus the AutoPi sensor readings for validation of speed and acceleration measurements. In the case of validating energy consumption measurements, the accumulated energy from wheel torque and instant traction power readings is compared to the total measured trip consumption.

In Figure 3a, the CAN bus speed is compared to the speed computed from the GPS signal. Due to the noise in the GPS signal (see Table 3), the signal was smoothed for visualization purposes. Figure 3b shows the vehicle distance traveled calculated from the CAN bus speed versus the distance traveled calculated from the GPS speed. In Figure 3c, the CAN bus longitudinal acceleration is compared to the AutoPi longitudinal acceleration. In Figure 3d, the cumulative sum of instant energy consumption calculated from the CAN wheel torque and traction power sensor is compared to the total measured trip energy consumption.

It is observed from Figure 3a that the CAN speed corresponds well with the GPS speed. The mean absolute error (MAE) between CAN speed and raw GPS speed is 5.44 km/h, and the Pearson correlation coefficient, *r*, is 0.907. It is also found that the difference between calculated distances is small (see Figure 3b), with an MAE of 6.5 m and *r* of 0.999. It is observed from Figure 3c that there is a near-perfect match between accelerometers. The MAE is 0.095 m/s^2^ and the *r* is 0.983. Finally, it is observed from Figure 3d that the cumulative sum of instant energy consumption calculated from the CAN wheel torque and traction power sensor resembles the total measured trip energy. It is also found that the total trip energy is reported in steps/resolution of 1 kWh. Hence, the instant traction energy from the CAN wheel torque sensor is further utilized in this study to achieve the highest possible sampling rate and resolution (see Table 3).

### 5.2. Field Measurements

Two roads in the LiRA project were selected for further investigation and demonstration of the proposed method; highway M3 in the southbound direction and urban ring road O2 (both directions) in the proximity of Copenhagen city, Denmark. Figure 4 depicts a map of the two roads; highway M3 is shown as a black dashed line, and ring road O2 is shown as a black dashed-dotted line. The choice of roads was guided by the desire to cover a wide variety of pavement conditions w.r.t age, type, and distress severity. They represent realistic scenarios for a wide-area road monitoring system.

Road roughness data were measured on 10 September 2020. The data were collected with a P79 profilometer operated by the Danish Road Directorate. The P79 is a van equipped with a high-quality GPS and a beam hosting 25-point lasers. It delivers longitudinal and transverse profiles at 0.1 m intervals. The P79 vehicle also measures the road slope utilized in this study. Figure 5 depicts the standard data from the P79 for the two roads shown as a black dashed line and a dash-dotted line for the highway road and the urban road (respectively). The IRI is plotted here to give an ‘intuition’ of the roughness variability, which is expected to cause variability in energy consumption.

The vehicle data were collected in the autumn of 2020 and spring of 2021 utilizing seven different GM cars. Designated drivers were instructed to drive in the right lane at a constant speed of 90 km/h on the highway road and 50 km/h on the urban road or follow the speed limit (e.g., in case this was lower than the instructed speed) or the traffic (e.g., in the case of congestion).

An overview of the timing of measurement campaigns, vehicles used, and surface conditions are shown in Table 5; the heading ‘Unit’ refers to the different AutoPi units (cars) used.

Once the GM car data were transmitted and stored in a database, several processing steps were carried out: (i) re-orientation of accelerometer axes to align with the principal vehicle axes; (ii) smoothing—removing digital errors in time series signals; (iii) map-matching—where the GPS roads are corrected using the P79 measurements; (iv) interpolation—where GPS coordinates are assigned to all sensor readings; and (v) structuring of data—where sensor readings are resampled to ensure consistency between EVs—and reference data across all sensors.

For a given car pass, time-series data shared via the CAN bus included: vehicle speed, road slope, vehicle acceleration in the travel direction, and instantaneous traction force. This time series was first resampled at 50 Hz. Then, the speed data were integrated w.r.t. time to provide cumulative distance and the traction energy was calculated from Equation (Equation 7). Lastly, all measured and calculated values were resampled based on 1 m distance intervals to produce a dataset with *v*—vehicle speed [km/h], β—road slope [m/km], *a*—longitudinal acceleration [m/s^2^] and $E_{10}$—average traction energy per 10 m [Wh]. An electric car’s typical average total traction energy, $E_{10}^{m}$, is 1.5 Wh per 10 m.

Average vehicle speed and measured traction energy from the GM cars traveling over the same road are visualized in Figure 6. The speed is shown as a black dashed line and dash-dotted line for the highway road and the urban road (respectively), and the traction energy as a grey dashed line and dashed-dotted line for the highway road and the urban road (respectively).

It is observed from Figure 6 that cars traveling over the urban road experienced more variability in energy consumption compared to vehicles traveling over the highway. This result is mainly due to the difference in speed variations; the speed on the highway is relatively high and constant, whereas the car speed on the urban road is lower and relatively variable. It may also be seen that sometimes the car speed drops to almost zero; these situations, which mostly correspond to light crossings, are not expected to provide information about the road condition [58]. Overview of key characteristics of the selected roads employed in this study are summarized in Table 6; IRI_10_ is the 10 m moving average of the IRI reported by the P79 profilometer. Moreover, μ is the section mean, and σ is the section standard deviation.

It can be seen from Table 6 that the average speed for the highway is higher compared to the urban road. It is also seen that the standard deviation in speed between single passes is high for the highway. In contrast, the standard deviation in speed for the urban road is relatively constant. The different road types/categories can explain this. The speed on a highway is mainly affected by traffic flow. Thus, phenomena such as congestion during rush hours may cause significant variability in vehicle speed. The speed is mainly controlled by road geometry, such as light crossings on urban roads. The table also shows that the energy consumption is slightly higher for the highway compared to the urban road and that the energy consumption increases with increasing average speed. It is also seen that the standard deviation in traction energy consumption is higher on the urban road compared to the highway. Finally, it is seen that the standard deviation in traction energy consumption increases with the increasing standard deviation in speed for the highway road.

## 6. Investigation

### 6.1. Validation of Normalization Technique

In order to visualize the influence of the normalization technique and validate the proposed method, the road slope versus measured and normalized energy data are plotted in Figure 7. The road slope from the P79 is utilized to minimize the influence of measurement errors.

For road sections where the car drives at a constant speed, the energy consumption must vary according to the longitudinal slope since all other driving resistances are constant, as verified/shown in Figure 7a. The chart presents $E_{10}$ versus β for two different vehicles driving at a constant speed (i.e., highway ‘pass no. 2’ and ‘pass no. 10’): 23.9 m/s (86 km/h) and 29.4 m/s (106 km/h). The chart in Figure 7b presents the traction energy normalized for acceleration $E_{10}^{n,a}$ versus β for the same data, i.e.,
(8)$$E_{10}^{n,a}=E_{10}^{m}-E_{10}^{a}$$

The chart in Figure 7c presents the traction energy normalized for acceleration and speed $E_{10}^{n,a,s}$ versus β for the same data, i.e.,
(9)$$E_{10}^{n,a,v}=E_{10}^{m}-E_{10}^{a}-E_{10}^{D}-E_{10}^{R}$$

Finally, the chart in Figure 7d presents the traction energy normalized for acceleration and speed $E_{10}^{n,a,v}$ versus β for two different car passes on Highway M3 with a standard deviation in speed σ(v) of 0 and 21 km/h, respectively (i.e., with low and high variability in speed). For each dataset, a black dashed and dashed-dotted regression line is included.

From Figure 7a, it is observed that the slope of the regression lines is almost identical and parallel, increasing with increasing positive β. The data are characterized by some noise with a moderate to a high positive correlation. The slope of the lines is 0.028–0.036, and the Pearson correlation coefficient, *r* is 0.68–0.83. Since it is impossible to drive at an exact speed (i.e., 86 km/h and 106 km/h) there is some noise in the data caused by small changes in longitudinal acceleration. Figure 7b shows that normalization w.r.t acceleration results in reduced noise and increased correlation. The regression lines are parallel and the offset is relatively unchanged. The slope of the lines is 0.044–0.046, and *r* is 0.93–0.96. Figure 7c shows that normalization w.r.t speed results in a change in offset—the regression lines almost coincide. The slope of the regression lines and the noise in the data are relatively unchanged. The same trend is observed for full trips with high variability in speed, as shown in Figure 7d. The noise in the data for the trip with high variability in speed is higher than the noise in the data for the trip with low variability in speed. However, both datasets are characterized by a high correlation. The slope of the lines is 0.045–0.048, and the *r* is 0.91–0.97.

The spikes in the data around a slope of zero in Figure 7b,c for ‘pass no. 10’, between a slope of −30 and −10 in Figure 7d for ‘pass no. 6’ correspond to situations with large variations in acceleration. Thus, energy data contain some information about acceleration after normalization. However, the approach seems robust; the additional ‘noise’ does not result in biased data (i.e., the results are very consistent for the two significantly different speed profiles).

The results in Figure 7 can also be used to ensure that the value magnitudes of data have physical meaning. From Figure 7d, the average slope of the regression lines gives the expression $E_{10}^{\beta }$ = 0.0465 β. Inserting the expression in Equations (Equation 4) and (Equation 7) (see Table 4) yields
(10)$$mg\beta \Delta x=0.0465\beta \times 3600\times 1000\to m=\frac{0.0465\times 3600\times 1000}{g\Delta x}$$

Utilizing *g* = 9.81 m/s^2^ (earth’s gravity acceleration), Δx = 10 m (length of analysis section), the resulting vehicle mass, *m*, is 1706 kg, which falls within the expected range of 1500–1900 kg (see Table 2). Factors 3600 and 1000 were included for converting the energy from Watt hours [Wh] to Newton meters [Nm] and slope from meters per kilometer [m/km] to meters per meter [m/m] (respectively).

Figure 8 presents the traction energy consumption results from a single pass on Highway M3. Figure 8a depicts the measured and predicted traction energy consumption along Highway M3. Figure 8b shows the influence of the normalization for the same dataset.

Figure 8a shows that the predicted traction energy consumption resembles the measured traction energy consumption; the Pearson correlation coefficient is 0.966. Normalization for slope and acceleration significantly influences the shape of the traction energy consumption curves, as shown in Figure 8b; $E_{10}^{n,a}$ resembles the $E_{10}^{n}$ curve. As expected, the speed and slope normalization primarily cause a shift in the curves since these parameters are relatively constant over longer road sections.

### 6.2. Estimating Road Energy Efficiency

This section investigates the link between normalized energy consumption and road pavement roughness. The analysis is divided into two steps. First, the collected data are aggregated and divided into road groups and roughness categories to discover overall patterns in the data. Next, the method’s reproducibility and performance are tested for individual car passes and shorter road sections.

The Pearson correlation coefficient, *r*, is used to quantify the relationship between energy and road roughness. The method’s reproducibility is quantified from the standard deviation, σ(r). In order to test the statistical significance of the results, one-way Analysis of Variance (ANOVA) tests are carried out [59]. ANOVA tests the hypothesis that the means for each category/group of data samples is equal, also referred to as the ‘null hypothesis’ ($H_{0}$), against the ‘alternative hypothesis’ ($H_{1}$) that the means are not equal. The test returns the *p*-value. A low *p*-value (typically below 0.05) indicates that the analysis results are significant.

From Table 6, it is observed that the ratio between roughness for the urban road and the highway road is 2.55. The corresponding ratio in normalized energy consumption is 2.95. The ratios in standard deviation are 3.1 and 1.8 for ${\mathrm{IRI}}_{10}$ and $E_{10}^{n}$ (respectively). These observations indicate that the normalized energy holds some information about the road roughness, e.g., the method can capture variations in energy consumption due to large variations in road roughness.

In order to test the hypothesis, the normalized energy data are aggregated for 15 km of road and then divided into three groups. Group no. 1 contains data from both the highway road and the urban road, group no. 2 contains data from the highway road, and group no. 3 the data from the urban road. The data are divided into five categories within each group, each representing roughness properties from very smooth (i.e., low) to very rough (i.e., high). Each category contains 20 percent of the data, i.e., from the 0–20th percentile to the 80th–100th percentile for ‘very smooth’ to ‘very rough’ (‘respectively’). Next, the linear correlation between means in each group is calculated as shown in Figure 9; the mean value in each group is shown as a blue square marker. For each dataset, a red-dashed regression line is included.

It is observed from Figure 9 that there is a strong linear relationship between $\mu ({\mathrm{IRI}}_{10})$ and $\mu (E_{10}^{n})$, for all three groups. The correlation coefficients are 0.92, 0.97, and 0.75 for groups no. 1, 2, and 3 (respectively). Utilizing the trend line in Figure 9a, it is found that a one unit increase in $\mu ({\mathrm{IRI}}_{10})$ results in a 3.4% increase in $\mu (E_{10}^{n})$.

The ANOVA test results for group no. 1 are visualized in the box plot in Figure 10. The box plot shows the data distribution within each road roughness category, i.e., ‘very rough,’ ‘rough,’ ‘medium rough,’ ‘smooth,’ and ‘very smooth’. On each box, the red line halfway mark indicates the median, and the bottom and top edges of the box indicate the 25th and 75th percentiles (respectively). The whiskers extend to the most extreme data points not considered outliers, and the outliers are plotted individually using a red cross marker.

The results are summarized in Table 7; it is observed that a *p*-value of zero is obtained for all three groups, indicating that differences between column means are significant. The table shows the between-category variation (‘columns’) and within-category variation (‘error’). SS is the sum of squares, and df is the degrees of freedom. The total degree of freedom is the total number of observations minus one. The between-category degrees of freedom are the number of categories minus one. The within-category degrees of freedom are total degrees of freedom minus the between-category degrees of freedom. The *F*-statistic is the ratio of the mean squared errors. The *p*-value is the probability that the test statistic can take a value greater than the value of the computed test statistic. MS is the mean squared error (i.e., SS/df) for each source of variation (i.e., P(*F* > SS)). The total number of observations is 300,000 for group no. 1 (i.e., 1 point per meter for ten car passes on 30 km of highway and urban road) and 150,000 for groups no. 2 and 3 (i.e., 1 point per meter for ten car passes on either 15 km of highway or urban road).

Data from individual car passes are evaluated in windows of 100, 250, 500, 1000, and 2500 m to test the method’s reproducibility and performance for shorter road sections. The choice of window size is guided by the following: (i) standard road condition parameters are typically reported in 10 to 100-m intervals; (ii) the measured energy consumption is constant over more extended road sections compared to traditional standard road condition parameters; and (iii) to ensure overlap between time series data collected from different sources at different speeds (i.e., GM cars and P79 vehicle).

Figure 11a,b present the results from a single pass utilizing a window size of 1000 m for the highway road and urban road (respectively). The measured roughness data utilizing standard methods is shown as a black dashed line, and the measured normalized energy data from the GM cars as a solid grey line.

It is observed from Figure 11 that the normalized energy resembles the measured road roughness. The correlation coefficient is 0.11–0.52 for the highway road and 0.13–0.61 for the urban road. The method performs slightly better for urban roads than highways, probably due to the more significant variability in road conditions and less variability in speed between individual passes. There are also lags/delays between the two data series on some sections of the roads. This may be a result of actual differences in the parameters measured. However, it could also result from vehicle location errors, e.g., noisy GPS signals or speed variations. An overview of the correlation coefficients for all passes and window sizes is summarized in Table 8. The Table also shows the average correlation coefficient, μ(r), the standard deviation of the correlation coefficient, σ(r), as well as the Coefficient of Variation (CoV), for each window size.

It is observed from Table 8 that a low to moderate positive correlation coefficient is obtained for all car passes and all window sizes on both the highway road and urban road. The magnitude differences between the highway and urban road results are small. The average correlation coefficients are 0.15–0.37 and 0.16–0.42, and the standard deviations are 0.06–0.21 and 0.09–0.19 for highway and urban roads (respectively). The mean and standard deviation of the correlation coefficients increase with the window size. The minimum CoV is found for the window size of approximately 1000 m. Finally, it is observed that the results are statistically significant; the *p*-value is 0.07–0.39 $\times \;10^{-2}$ and 0.09–0.95 $\times \;10^{-2}$ for the highway road and urban road (respectively).

The *p*-values are obtained from a manufactured dataset. First, a synthetic set of normalized energy consumption data were generated for each car pass. This was done, assuming a normal distribution of data, with a mean and standard deviation equal to the corresponding data for the real car pass. Next, the manufactured dataset was utilized to produce a set of synthetic correlation coefficients. Finally, the *p*-values were calculated from an ANOVA test. The null hypothesis, in this case, is that the correlation coefficients in Table 8 are random, with a mean of zero, and therefore not statistically different from correlation coefficients produced by the synthetic/manufactured dataset.

The results of the ANOVA test for a window size of 1000 m are visualized in the box plot in Figure 12a,b for the highway road and urban road (respectively); the correlation analysis for the proposed method is named ‘model’ and the manufactured randomly generated correlation coefficient is named ‘random’.

Figure 12a,b shows the correlation coefficients produced by the ‘random’ model range from approximately −0.15 to 0.15 and −0.2 to 0.3 for highways and urban roads (respectively). The means of both ‘random’ correlation coefficients are around zero, whereas the means of the proposed model are 0.32 and 0.37 for highways and urban roads (respectively). Thus, the means of the ‘model’ and ‘random’ correlation coefficients differ significantly, resulting in low *p*-values (see Table 8).

### 6.3. Mapping and Visualization

The outcome of the proposed concept is visualized in Figure 13 and Figure 14. In Figure 13a,b, the total and normalized energy consumption for the highway and the urban road is visualized on a map (respectively). The energy map may give users/operators a quick overview of the total energy consumption on the road network. The information can be used to identify critical areas of the road infrastructure, e.g., w.r.t the physical condition of the road, as shown in the example in Figure 13c,d.

Figure 13a shows that the energy consumption for the highway road is relatively constant over more extended periods compared to the urban road. This is expected since the speed on highways is relatively constant, whereas the car speed in an urban environment changes more frequently (e.g., due to congestion and light crossings). It is also shown that the peak energy consumption is higher on the highway compared to the urban road. Figure 13b shows that the normalized energy consumption is higher on the urban road compared to the highway road. Figure 13c,d show a close-up of the same dataset enabling users to study energy variations on short road sections.

Further, the concept allows for more detailed analysis, e.g., analysis of individual passes and energy components, as exemplified by the bar plot in Figure 14; the total measured energy consumption is $E^{m}$, and the predicted energy consumption values from the road slope, acceleration, aerodynamic drag, and tire rolling are $E^{\beta }$, $E^{a}$, $E^{D}$, and $E^{R}$ (respectively). The color bars represent energy data from the ten individual car passes. Moreover, shown is the mean, μ(v), and standard deviation, σ(v), speed for each car pass. Figure 14a,b show the total energy consumed and its components over 25 km of highway and urban road (respectively).

It is observed from Figure 14a that increasing average speed results in increased energy consumption. This can be explained by the increase in aerodynamic drag and tire rolling. Comparing Figure 14a,b, it is found that the total energy consumption is higher for the highway road compared to the urban roads. Moreover, it is observed that the average speed for the urban roads is almost constant (over longer sections) for all cars. It is also found that energy consumption is less affected by speed and more affected by acceleration compared to the highway road. This can be explained by the lower speed in an urban environment, resulting in significantly decreased aerodynamic drag, while simultaneously, the frequency of acceleration and deceleration events increases.

## 7. Summary and Conclusions

In this study, a new road energy efficiency monitoring system was proposed. The system was based on measurements from in-vehicle sensors collected onboard with an Internet-of-Things (IoT) device. The data were transmitted periodically before being processed, normalized, and saved in a database. The normalized energy was then linked to the road pavement roughness and visualized on a map.

This is the first time such a concept has been utilized to quantify changes in vehicle energy consumption caused by pavement roughness on highways and urban roads. The approach has the advantage that it enables analysis of data across vehicle types in a real-scale setting and is, therefore, superior to other methods.

The new method was first validated utilizing a limited dataset of vehicles driving on a highway road. The results from this verification effort show that energy data can be normalized and that the physical models proposed apply to the problem.

Experimental data from ten nominally identical electric cars driven over 25 km of highways and urban roads were utilized to investigate the relationship between normalized energy consumption and road pavement roughness.

As a first step, the normalized energy data were aggregated into three road groups/ classes. Within each group, the data were further subdivided into five road roughness categories, each containing 20% of the group data. The results showed a strong linear relationship between the normalized energy consumption and road roughness. Then, the relationship between normalized energy and road roughness for individual passes and the methods’ reproducibility were assessed. In this context, the data were analyzed in 100- to 2500-m windows. It was found that the normalized energy consumption collected from individual cars resembled the measured road roughness. The results also showed a low-to-moderate positive linear relationship for all car passes and window sizes on both highways and urban roads.

Analysis of Variance (ANOVA) tests showed that the results obtained are statistically significant for both aggregated data and data from individual passes. Thus, it is concluded that the normalized energy consumption holds some information about the physical condition of the road.

The method enables road network mapping of energy data. Such energy consumption maps give users/operators a quick overview of the total energy consumption on the road network and help identify critical areas of the road infrastructure. The method also enables energy data analysis in terms of physical phenomena (e.g., road slope, acceleration, speed, tire-rolling, or road condition) and source (e.g., vehicle type or pass), as well as how these parameters evolve.

The main findings from this study can be summarized as follows; (i) a new method for estimating road energy efficiency was proposed and successfully utilized to analyze energy data on highways and urban roads; (ii) the average normalized energy consumption is 0.13 and 0.37 Wh per 10 m for highways and urban roads (respectively); (iii) the normalized energy consumption is positively correlated to surface roughness—the average correlation coefficient is 0.88 for aggregated data and 0.32 and 0.39 for 1000-m road sections on highways and urban roads (respectively), and (iv) an increase in IRI of 1 m/km results in a 3.4% increase in normalized energy consumption.

Considering the emergence of connected vehicle technologies, the method seems promising and can potentially be used as a platform for future large-scale road energy efficiency monitoring. The normalized energy consumption is a new pavement condition indicator that supports decision-making and may contribute to improved pavement management. Additionally, the normalized energy can be applied to initiatives to create labeling systems for road infrastructure similar to those used in consumer sectors. In this context, it is envisioned that the proposed concept will be combined with other important factors affecting vehicle energy consumption, such as road pavement type, road classification, road geometry, traffic information, and weather conditions. Hence, this issue is of critical importance and requires further research.

## Figures and Tables

**Figure 1 sensors-23-02756-f001:**
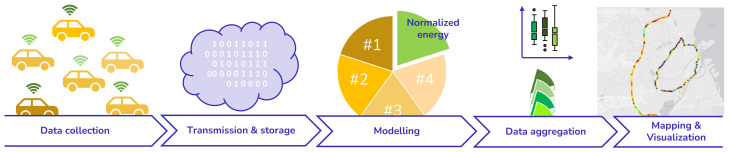
Proposed framework for road energy efficiency monitoring, showing the data collection from vehicle sensors, data transmission, storage, normalization of data utilizing physical models, data aggregation, and mapping and visualization.

**Figure 2 sensors-23-02756-f002:**
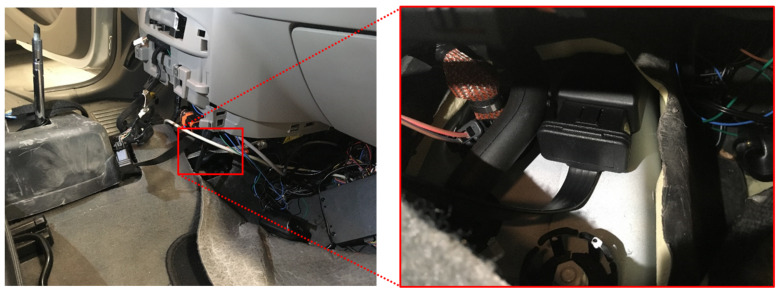
Picture of the passenger side of the Renault Zoe with the access panel to the middle console removed and the location of the AutoPi telematics unit inside the middle console.

**Figure 3 sensors-23-02756-f003:**
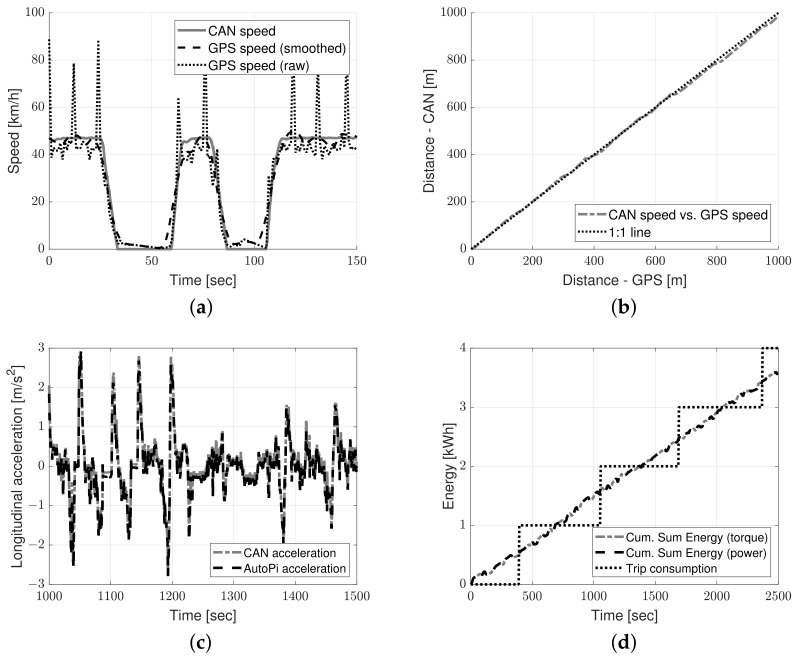
Validation of data collection platform: (**a**) speed from Can bus versus speed calculated from AutoPi GPS sensor, (**b**) distance calculated from CAN bus speed versus distance calculated from GPS speed, (**c**), CAN bus acceleration versus AutoPi acceleration, and (**d**) cumulative sum of instant energy consumption calculated versus the total measured trip energy consumption.

**Figure 4 sensors-23-02756-f004:**
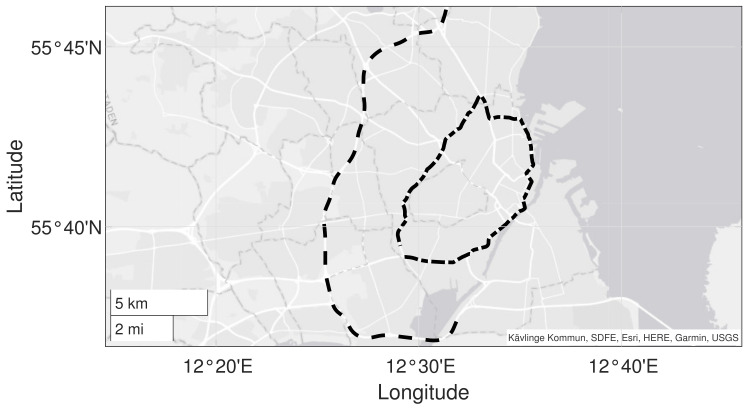
Overview of roads utilized in the experimental investigation: map of highway road M3 (marked with a black dashed line) and urban city ring O2 (marked with black dashed-dotted line) in the proximity of Copenhagen city, Denmark.

**Figure 5 sensors-23-02756-f005:**
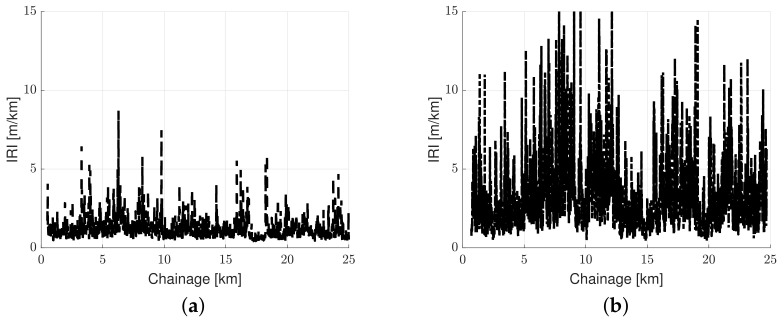
Overview of standard road condition data utilized in the experimental investigation: (**a**) measured IRI (black line) for highway road M3 in the southbound direction and (**b**) measured IRI for urban road O2.

**Figure 6 sensors-23-02756-f006:**
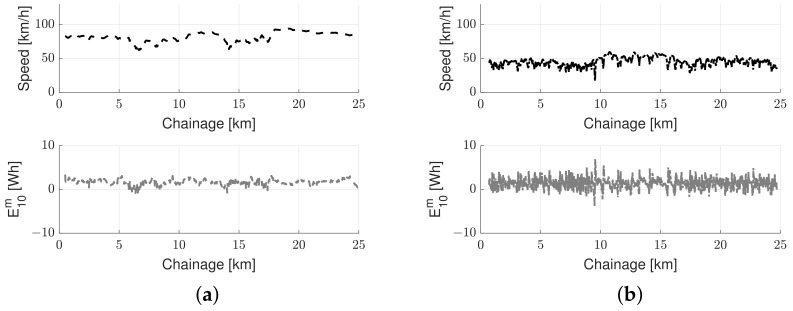
Typical data utilized in the experimental investigation: measured car speed (black line) and energy consumption (gray line) for (**a**) highways and (**b**) urban roads.

**Figure 7 sensors-23-02756-f007:**
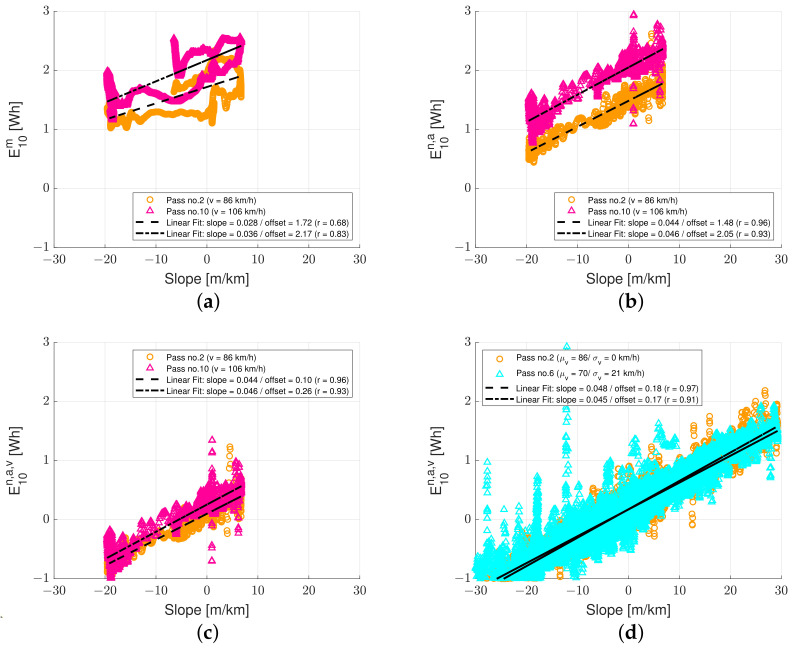
Slope versus traction energy consumption: (**a**) slope versus measured traction energy at constant speed, (**b**) slope versus traction energy normalized for acceleration at a constant speed, (**c**) slope versus traction energy normalized for speed and acceleration at a constant speed and (**d**) slope versus traction energy normalized for speed and acceleration for varying speeds (full trip).

**Figure 8 sensors-23-02756-f008:**
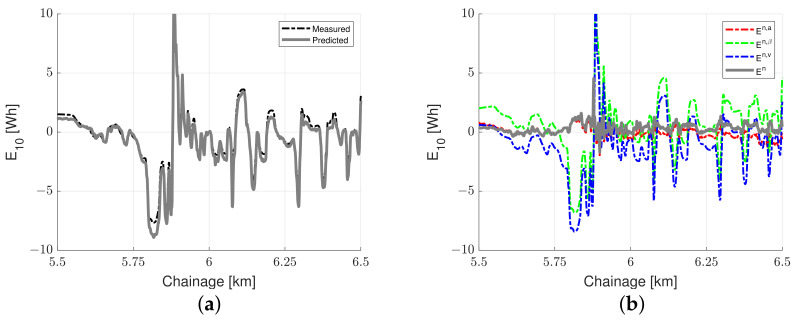
Energy consumption data for a single trip on M3: (**a**) measured versus predicted energy consumption and (**b**) energy normalization.

**Figure 9 sensors-23-02756-f009:**
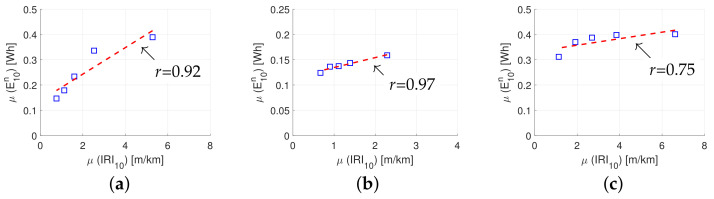
Relationship between road roughness and normalized energy for (**a**) group no. 1: highway and urban road, (**b**) group no. 2: highway road, and (**c**) group no. 3: urban road.

**Figure 10 sensors-23-02756-f010:**
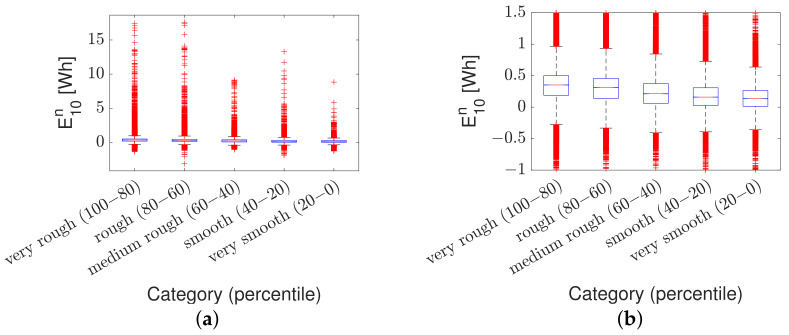
Box-plot of aggregated normalized energy data divided into road roughness categories: (**a**) energy data for group no. 1 and (**b**) close-up of the same data.

**Figure 11 sensors-23-02756-f011:**
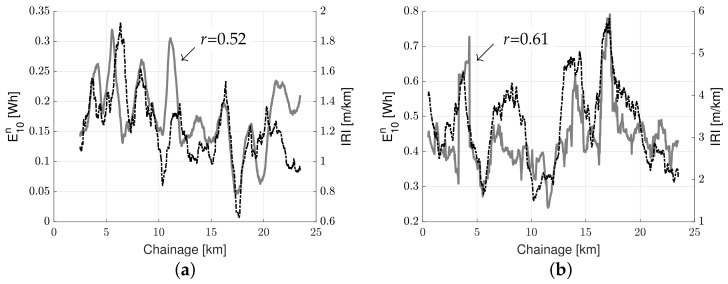
Normalized energy versus road roughness utilizing a 1000-m moving average window: (**a**) energy highway road pass no. 2 and (**b**) urban road pass no 7.

**Figure 12 sensors-23-02756-f012:**
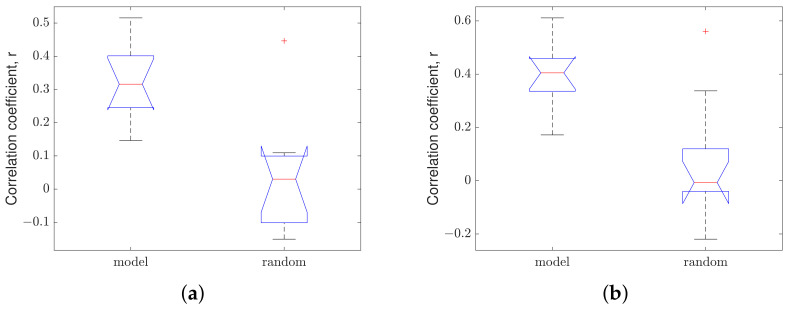
Box-plot of measured and randomly generated correlation coefficients for 10 car passes: (**a**) highway road and (**b**) urban road.

**Figure 13 sensors-23-02756-f013:**
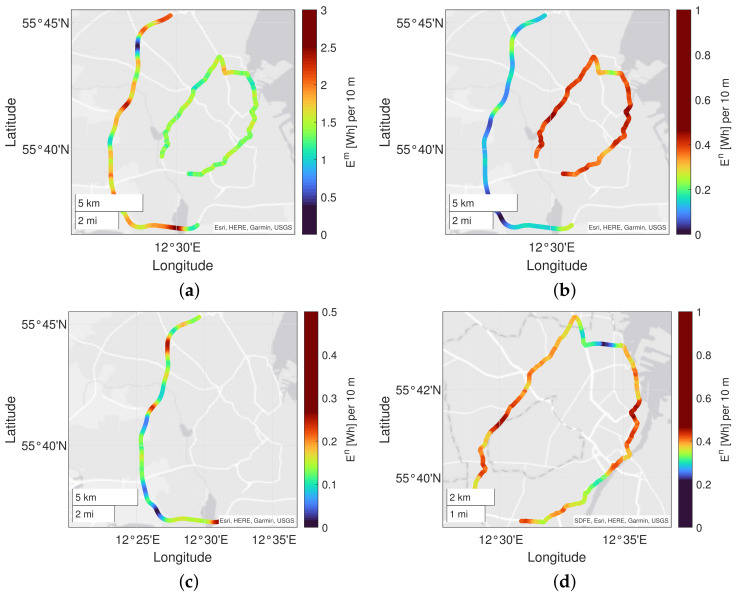
Mapping and visualization of average energy consumption data from ten car passes on highway road M3 and urban ring road O2: (**a**) total traction energy consumption, (**b**) normalized energy consumption, (**c**) normalized energy consumption on the highway road and (**d**) normalized energy consumption on the urban road.

**Figure 14 sensors-23-02756-f014:**
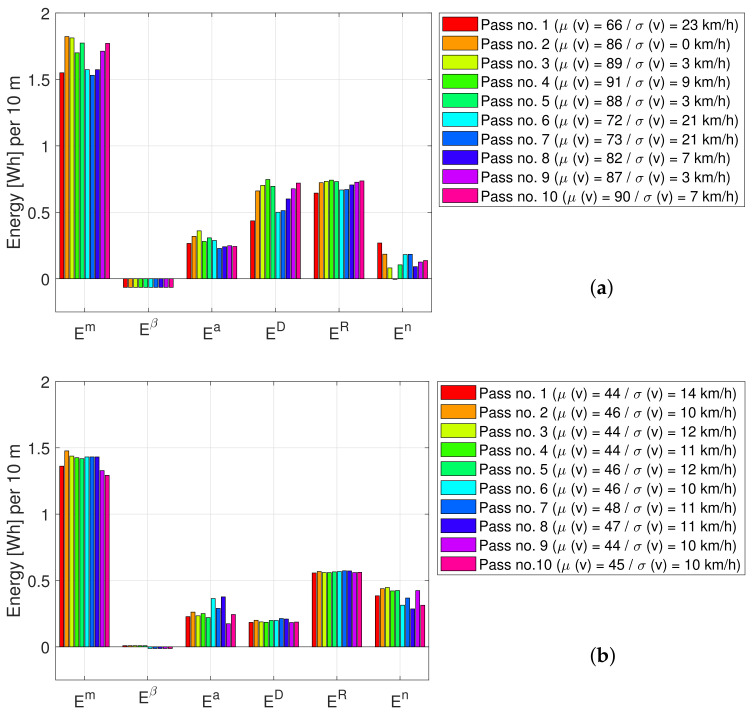
Bar plot of energy consumption data split into car passes and energy components: (**a**) highway road M3 and (**b**) urban ring road O2. Each car pass is shown as a single bar with a different color.

**Table 1 sensors-23-02756-t001:** Summary of related research work.

Method	Reference	Limitation
Empirical	[36,37,38,39,40]	(i) Model complexity and correctness
		(ii) Calibration required
		(iii) Complexity of simulation (not directly measured)
Mechanistic	[43,44,45,46,47,48]	(i) Model correctness
		(ii) Extensive system characterization may be required
		(iii) Complexity of simulation (not directly measured)
Data-driven	[50]	(i) Vehicle types considered (heavy logging trucks)
		(ii) Road class (low-volume rural roads)
		(iii) Road pavement type (gravel)
		(iv) Effect of acceleration not considered

**Table 2 sensors-23-02756-t002:** Specification/metadata for Renault Zoe EVs utilized in this study.

Parameter	Value	Unit
Curb weight, $m_{c}$	1480	kg
Gross weight, $m_{g}$	1966	kg
Front area of the vehicle	2.33	m^2^
Drag coefficient, $C_{d}$	0.29	–
Torque	220	Nm
Power	68	kW
Effective wheel radius [56]	≈300	mm

**Table 3 sensors-23-02756-t003:** Description of EV signals utilized in the present study.

Signal	Unit	Source	Sampling Rate	Resolution/Accuracy
Vehicle location/GPS	DD	AutoPi	1 Hz	±10 m
Longitudinal acceleration	m/s^2^	AutoPi	50 Hz	0.01 m/s^2^
Longitudinal acceleration, *a*	m/s^2^	CAN bus	100 Hz	0.01 m/s^2^
Wheel torque, $T_{whl}$	Nm	CAN bus	100 Hz	0.01 Nm
Braking torque, $T_{brk}$	Nm	CAN bus	100 Hz	0.01 Nm
Vehicle speed, *v*	km/h	CAN bus	50 Hz	0.01 km/h
Traction power, $P_{tr}$	W	CAN bus	10 Hz	0.01 W
Trip consumption, $E_{trip}$	kWh	CAN bus	2 Hz	1.00 kWh

**Table 4 sensors-23-02756-t004:** Overview of the relevant energy components utilized in the present study.

	Measured	Normalized	Drag	Tire	Climbing	Acceleration
Force:	$F_{tr}^{m}=\frac{T_{whl}}{r_{t}}$	$F_{tr}^{n}$	$F^{D}$	$F^{R}$	$F^{\beta }$	$F^{a}$
Energy:	$E_{\Delta x}^{m}$	$E_{\Delta x}^{n}$	$E_{\Delta x}^{D}$	$E_{\Delta x}^{R}$	$E_{\Delta x}^{\beta }$	$E_{\Delta x}^{a}$

**Table 5 sensors-23-02756-t005:** Overview of measurement campaigns.

Road	TaskID	Unit	Date	Time		Temperature	Surface
	/Trip			Start	Stop	[°C]	Conditions
M3	5642	TMU2	3 November 2020	14:24	17:06	12	Dry
	7567	TMU11	16 April 2021	18:08	18:58	11	Dry
	7885	TMU11	24 April 2021	09:07	10:51	7	Dry
	7895	TMU11	24 April 2021	12:38	14:20	9	Dry
	7995	TMU10	26 April 2021	11:49	16:52	6	Dry
	8189	TMU12	30 April 2021	09:30	11:38	10	Dry
	Profilometer (P79)		10 September 2020	18:52	19:16	16	Dry
O2	8040	TMU14	27 April 2021	15:58	17:48	11	Dry
	8227	TMU11	30 April 2021	18:19	20:40	9	Dry
	9289	TMU6	15 May 2021	06:57	08:47	9	Dry
	10218	TMU12	20 May 2021	18:40	20:21	13	Dry
	10900	TMU11	27 May 2021	07:14	11:31	11	Dry
	11360	TMU7	30 May 2021	07:28	09:04	13	Dry
	11367	TMU7	30 May 2021	09:04	12:42	16	Dry
	Profilometer (P79)		10 September 2020	10:04	11:29	15	Dry

**Table 6 sensors-23-02756-t006:** Overview of selected roads employed for demonstration of the road energy efficiency monitoring system.

Road	Length	$\mu ({\mathrm{IRI}}_{10})$	$\sigma ({\mathrm{IRI}}_{10})$	Pass	TaskID	$\mu_{v}$	$\sigma_{v}$	$\mu (E_{10}^{m})$	$\sigma (E_{10}^{m})$
	[km]	[m/km]	[m/km]	[No.]	/Trip	[km/h]	[km/h]	[Wh]	[Wh]
M3	25	1.27	0.67	1	5642	63.12	23.31	1.49	1.70
				2	7567	86.27	0.08	1.79	0.50
				3	7885	89.43	3.33	1.79	0.54
				4	7885	91.18	9.52	1.68	1.35
				5	7895	88.24	2.88	1.73	0.67
				6	7995	69.89	21.10	1.53	1.61
				7	7995	70.78	21.44	1.47	1.55
				8	7995	82.21	7.18	1.54	1.04
				9	7995	87.16	3.12	1.67	0.66
				10	8189	91.22	6.41	1.76	0.74
O2	25	3.24	2.09	1	8040	47.17	9.46	1.41	2.48
				2	8227	47.70	11.11	1.39	2.79
				3	8227	48.34	10.73	1.40	2.87
				4	9289	45.54	10.19	1.30	2.30
				5	10218	45.06	9.69	1.26	2.78
				6	10900	46.26	11.57	1.37	2.50
				7	11360	48.43	9.85	1.49	2.45
				8	11367	46.47	12.00	1.47	2.59
				9	11367	45.78	11.06	1.44	2.48
				10	11367	48.69	11.97	1.45	2.28

**Table 7 sensors-23-02756-t007:** ANOVA table for normalized energy divided into road groups and roughness categories.

Group	Road	Source	SS	df	MS	*F*	*p*-Value
1	Highway and Urban	Columns	2552.8	4	638.2	4089.0	0
		Error	46,821.7	299,996	0.16		
		Total	49,374.5	300,000			
2	Highway	Columns	19.2	4	4.8	91.2	0
		Error	7898.6	149,996	0.05		
		Total	7917.8	150,000			
3	Urban	Columns	162.1	4	40.5	163.4	0
		Error	37,202.5	149,996	0.25		
		Total	37,364.6	150,000			

**Table 8 sensors-23-02756-t008:** Summary of correlations between normalized energy consumption and road roughness for individual vehicle passes.

Road	Pass			Window Size			$\mu (E_{10}^{n})$	$\sigma (E_{10}^{n})$
		100 m	250 m	500 m	1000 m	2500 m		
	[No.]		Correlation Coefficient (*r*)				[Wh]	[Wh]
M3	1	0.04	0.11	0.14	0.15	0.01	0.28	0.24
	2	0.25	0.33	0.42	0.52	0.66	0.18	0.14
	3	0.20	0.27	0.32	0.35	0.32	0.08	0.18
	4	0.17	0.23	0.24	0.25	0.28	-0.02	0.23
	5	0.20	0.27	0.36	0.42	0.54	0.09	0.19
	6	0.14	0.25	0.34	0.40	0.54	0.18	0.24
	7	0.13	0.23	0.28	0.37	0.55	0.18	0.22
	8	0.09	0.11	0.18	0.28	0.21	0.09	0.22
	9	0.10	0.13	0.17	0.28	0.44	0.10	0.16
	10	0.12	0.17	0.18	0.19	0.17	0.13	0.19
	μ(r)	0.15	0.21	0.26	0.32	0.37		
	σ(r)	0.06	0.08	0.09	0.11	0.21		
	CoV	0.40	0.38	0.35	0.34	0.57		
	*p*-value	0.07 $\times \;10^{-2}$	0.03 $\times \;10^{-2}$	0.04 $\times \;10^{-2}$	0.03 $\times \;10^{-2}$	0.39 $\times \;10^{-2}$		
O2	1	0.18	0.30	0.38	0.50	0.62	0.30	0.59
	2	0.09	0.22	0.29	0.35	0.39	0.35	0.57
	3	0.04	0.12	0.15	0.17	0.36	0.30	0.41
	4	0.07	0.16	0.22	0.23	0.17	0.40	0.39
	5	0.33	0.40	0.45	0.46	0.39	0.30	0.57
	6	0.18	0.25	0.34	0.46	0.67	0.37	0.39
	7	0.17	0.29	0.44	0.61	0.72	0.42	0.62
	8	0.22	0.29	0.36	0.45	0.49	0.45	0.53
	9	0.15	0.22	0.30	0.36	0.30	0.42	0.44
	10	0.21	0.24	0.27	0.34	0.21	0.43	0.35
	μ(r)	0.16	0.25	0.32	0.39	0.43		
	σ(r)	0.09	0.08	0.10	0.13	0.19		
	CoV	0.56	0.32	0.31	0.33	0.44		
	*p*-value	0.02 $\times \;10^{-2}$	0.01 $\times \;10^{-2}$	0.03 $\times \;10^{-2}$	0.09 $\times \;10^{-2}$	0.95 $\times \;10^{-2}$		

## Data Availability

Data are available at: https://dx.doi.org/10.11583/DTU.c.6337148. Accessed on 17 January 2023.

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
