# Peer review of "Internet-of-Things (IoT) Platform for Road Energy Efficiency Monitoring"

_sensors, 2023, doi:10.3390/s23052756_

Round 1

Reviewer 1 Report

- how can you reduce energy consumption using IoT on roads and how can you monitor it

The literature review. I do not find the literature review convincing. I think that the authors need to do a much better review. can be cited  A provably secure key transfer protocol for the food enabled social internet of vehicles based on a confidential computing environment

-Authors are suggested to include the contributions of the proposed work at the end of Introduction section.

- Few of the references included in the related work are very old.

-The abstract needs improvements in terms of adding results.

-It is not clear how this work improves existing approaches. Comparison with previous approaches in a table is needed to understand the contributions.

-Description about experimental study needs to be improved. Meanwhile, good to see more details about design principle of experimental study and procedure.

- What are the evaluations used for the verification of results?

- Major contribution was not clearly mentioned in the conclusion part.

Reviewer 2 Report

The authors reported an interesting system for road energy efficiency monitoring based on measurements from in-vehicle sensors. These measurements were collected onboard with an IoT device and periodically transmitted. Experimental results were measured for the validation of the proposed system. This manuscript can be enhanced by considering the following issues:

1.-The introduction section should add the main advantages of the proposed system for road energy efficiency monitoring compared to other systems reported in the literature.

2.-The equation (2) should be revised.

3.- What are the main limitations of the proposed system?

4.- The authors should include more discussions on the results shown in Figures 7, 11, and 12. 

5.- Figure 8 contains labels that intersect the results. These labels must not obstruct the results.

Round 2

Reviewer 2 Report

This manuscript version was improved based on the reviewer's comments.